



# Variability of Basal Meltwater Generation During Winter, Western Greenland Ice Sheet

Joel Harper[1], Toby Meierbachtol[1], Neil Humphrey[2], Jun Saito[2], Aidan Stansberry[2]

[1] Department of Geosciences
Univ. of Montana
Missoula, MT 59812
U.S.A.

[2] Geology and Geophysics
Univ. of Wyoming
Laramie, Wyoming 82071
U.S.A.

*Correspondence to*: Joel Harper (Joel@mso.umt.edu)

**Abstract.** Basal sliding in the ablation zone of the Greenland Ice Sheet is closely associated with water from surface melt introduced to the bed in summer, yet melting of basal ice also generates subglacial water year-round. Assessments of basal melt rely on modelling with results strongly dependent upon assumptions with poor observational constraint. Here we use surface and borehole measurements to investigate the generation and fate of basal meltwater in the ablation zone of Isunnguata Sermia basin,

Western Greenland. The observational data are used to constrain estimates of the heat and water balances, providing insights into subglacial hydrology during the winter months when surface melt is minimal or non-existent. Despite relatively slow ice flow speeds during winter, the basal meltwater generation from sliding friction remains many fold greater than that due to geothermal heat flux. A steady acceleration of ice flow over the winter period at our borehole sites can cause the rate of basal water generation to increase by up to 20%. Borehole measurements show high but steady basal water pressure, rather than monotonically increasing

pressure. Ice and groundwater sinks for water do not likely have sufficient capacity to accommodate the meltwater generated in winter. Analysis of basal cavity dynamics suggests that cavity opening associated with flow acceleration likely accommodates only a portion of the basal meltwater, implying a residual is routed to the terminus through a poorly connected drainage system. A forcing from cavity expansion at high pressure may explain observations of winter acceleration in Western Greenland.




## 1 Introduction


The flow speed of the outer flanks of Greenland Ice Sheet (GrIS) typically exhibits distinct seasonal (e.g., Zwally et al., 2002) and diurnal (e.g., Shepherd et al., 2009) variations that can be doubling to tripling. Both observational (e.g., Andrews et al., 2014) and modelling (e.g., Schoof, 2010) studies have established a close association between the speed changes and evolving subglacial hydrologic conditions forced by surface melt. Yet, despite little to no surface melt in the winter months, this period is when the

vast majority of the overall ice displacement occurs (Sole et al., 2013) due to continuous motion (albeit relatively slow) over about two thirds of the year.

Water is also introduced to the bed during the winter via basal ice melt driven by geothermal heat, sliding friction, and strain heating of the ice. The magnitude of basal ice melt in the western Greenland ablation zone has been assessed by models which are

heavily dependent upon assumptions about sliding speed (e.g., Brinkerhoff et al., 2011; Meierbachtol et al., 2015). These studies suggest ice melt rates are spatially variable and reach values on the order of 2 cm $a^{-1}$ along the outer flanks of the ablation zone. While the water generated from basal melt is trivially small in comparison to that derived from surface melt in summer, basal ice melt may become important during the winter months, particularly when considering the accumulation of basal meltwater over time and space.


Basal meltwater generated in winter is potentially routed into storage sinks in the ice, absorbed by the groundwater system, routed into a connected subglacial drainage system, and/or accommodated by expansion of basal water cavities. The pressure, availability, and distribution of water dictates potential groundwater recharge and exchanges between the bed and storage sinks in the overlying ice, and the connectivity and water flux conditions dictate throughput to the terminus for winter discharge. The winter configuration

of the drainage system and associated ice/bed coupling drive winter ice flow, and dictate the initial hydrologic conditions that are perturbed by spring surface melt and ultimately result in a spring acceleration of ice flow. Yet, little is known about bed conditions in winter due to the paucity of observations and the difficulty of addressing critical sub grid-scale processes with models.

This study examines the role of basal meltwater generation in subglacial drainage system characteristics in the ablation zone of

Western Greenland. We focus on the Isunnguata Sermia outlet basin and on the winter months when surface melt is minimal to non-existent. We analyse data that include surface motion, measured by GPS stations and satellite, and subglacial conditions measured by in situ sensors placed in a network of boreholes to the bed. The borehole data offer direct context and constraint on bed conditions, and facilitate measurement-based constraints on basal meltwater generation and its fate in hydrologic processes.

## 2 Methods

### 2.1 Sites and measurements

The Isunnguata Sermia (IS) basin is directly north of the well-studied Russell Glacier and K-transect (van de Wal et al., 2008) near the centre of the ~450 km long land terminating portion of Western Greenland (Fig. 1). Within the IS basin we have comprehensive in situ measurements collected in 32 boreholes drilled to the bed along a transect of sites extending 46 km inland from the outlet

glacier terminus. Boreholes were drilled with a hot water system (Meierbachtol et al., 2013) and vary from ~100 m deep near the ice margin to 830 m deep at a site 46 km from the terminus.

We present data from sites 27, 33, and 46 km from the terminus, but our analysis is primarily focused on data collected at the 33 km site. Here nine boreholes extending 640-688 m to the bed were instrumented. Five surface GPS stations yield three full years



of in situ observations of surface velocity, and borehole inclinometers provide a direct measure of the basal sliding component of ice motion (Maier et al., 2019). Meteorological variables were also measured at the site, including surface ice melt by sonic distance sensor and rain precipitation by tipping bucket (Hills et al., 2018). The data and analyses in sections below draw upon previously published work on the borehole transect addressing subsurface conditions, including ice temperature, bed framework, water pressure, and borehole partitioning of ice deformation/basal sliding speeds.


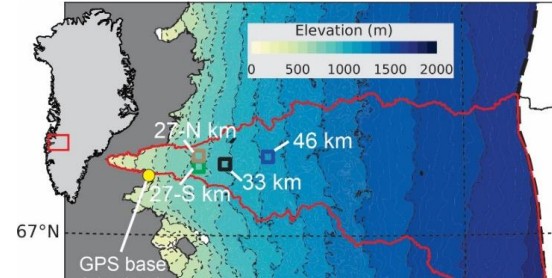

**Figure 1.** Map showing Isunnguata Sermia basin (red outline) in western Greenland. Inland basin boundary is clipped at the long-term position of the equilibrium line (see methods). Measurement sites for data shown here: borehole water pressures at 27N and 27S km; GPS speed at 46 km; basal water pressure, surface melt, and GPS speed at 33 km site.

Delineation of the IS basin catchment is based on the field of total potential of basal water, calculated under assumptions of ice density (910 kg m-3), and water pressure equal to ice overburden pressure (Wright et al., 2016). The catchment boundary was computed by the r.watershed module of GRASS GIS (GRASS Development Team, 2020). We impose an inland catchment

boundary defined by the long-term (1958-2016) equilibrium line altitude (ELA) from RACMO2.3p2 (Noël et al., 2018). The latter is not likely to be a true boundary with respect to basal water, but data limitations prevent analysis of conditions further inland, and previous work has suggested this boundary is a reasonable proxy for the onset of sliding conditions that control melt generation (Meierbachtol et al., 2016).

**2.2 Surface speed**

High temporal resolution details of winter velocity at the 33 km site are revealed by a three-year record of surface displacement measured by GPS. Five logging GPS stations installed in a diamond pattern, each with an antenna placed on a pole drilled into the ice, recorded data from summer 2014 to summer 2017. Trimble net-R9 or net-RS receivers with batteries and solar recharging were installed to run continuously during months with sunlight. The stations shut down for 1-2 months during the dark months of

December and January due to low power. The surface velocity was calculated from position data collected at 15 second intervals (Maier et al., 2019). Data were processed against a base station located off the ice sheet ~22 km away using the TRACK (version 1.29) differential kinematic processing module within GAMIT/GLOBK software (Herring et al., 2010). Here we focus on the central station, since the differences between stations are inconsequential to our results.

We assess the spatial pattern of the winter speed-up across the basin through analysis of the publicly archived MEaSUREs Version 2 monthly satellite velocity mosaics (Joughin et al., 2018a). The MEaSUREs product incorporates multiple SAR data sources, combined with Landsat feature tracking, and has a 200x200 m posting and 1-month temporal resolution (i.e., April values represent velocities calculated from displacements over the month of April). For our purposes of basin assessment with satellite data, we





define the winter period as October through the end of April. While this period conservatively captures winter conditions across

the basin, we note that winter weather conditions may initiate earlier and terminate later, depending on elevation and year.

The velocity products have intrinsic errors associated with the measurements and methodologies, as well as inconsistences in availability of the primary satellite data. The errors tend to be more problematic when considering finer detail and shorter time intervals (Derkacheva et al., 2020). Limitations of the velocity product thus dictate that our MEaSUREs data analysis is restricted

to the 2017-2018 winter, deemed to have the best quality (Joughin et al., 2018b). The average error over the basin of the velocity product for the month of October, 2017, is <3 m a$^{-1}$. The average velocity is <87 m a$^{-1}$ over the basin during that month, so the velocity error is ~3% of the average value. The greatest challenge is with differencing the velocity fields between the start and end of the winter period in order to examine basin-wide changes. We limit our results to show only values that rise above reported measurement errors. In other words, if differencing between start/end of winter products indicates a 20% speed change in a given

location, but errors terms alone yield a 15% change, we show results as a minimum change of 5%.

### 2.3 Basal water pressure

Boreholes fitted with water pressure transducers were used to directly measure basal water pressure. Data were collected at 5 to 15-minute intervals during summer and 15 to 30-minute intervals during winter. Details of the instrumentation are presented in

Wright et al. (2016). Pressure values were converted to fraction of ice overburden, with uncertainty in this calculation arising from transducer resolution, ice depth measurement, and assumed ice density, the latter being the greatest. Incorporating a range for ice density values from 900–917 kg m$^{-3}$ yields a total uncertainty of -0.022 to +0.026 times overburden (Wright et al., 2016). We present data from several boreholes drilled at the 33 km site, in addition to data from holes at 27 km-N and 27 km-S sites with records extending through the winter period.


### 2.4 Basal meltwater generation

### 2.4.1 Instrumented site

Our prior measurements in boreholes of the basal ice temperature and sliding speed allow the basal ice melt to be calculated with an unusually high level of observational constraint at our 33 km study site. Melt rate, $\dot{M}$, is calculated as

$$\rho L \dot{M} = \left( u_b \tau_b + Q_{geo} + K\left(\frac{\partial T}{\partial z}\right)\right),$$

(1)

where, $\rho L$ are ice density and latent heat of fusion respectively, $u_b$ is the borehole-measured basal velocity which is 96% of the surface speed (Maier et al., 2019), and $\tau_b$ is the calculated basal shear stress. The geothermal heat flux, $Q_{geo}$, is ~27 mW m$^{-2}$ and is constrained by direct measurements in a bedrock borehole located ~22 km away (Meierbachtol et al., 2015). Heat flow into the ice above the bed, $K\left(\frac{\partial T}{\partial z}\right)$, is known to be insignificant because our borehole temperature measurements indicate a temperate ice layer exists above the bed (Harrington et al., 2015; Hills et al., 2017).


The basal meltwater generation is a model result rather than a direct measurement at the bed. We adopt the common assumption that basal ice melt from sliding friction is a function of sliding speed and driving stress (Cuffey and Paterson, 2010), without higher order terms. Our borehole observations minimize uncertainties related to basal temperature and sliding speed, and provide an unusually accurate measure of ice depth. However, uncertainties related to driving stress remain. Previous work (Meierbachtol. et

al., 2016) examined the driving stress and uncertainties across the basin, including a profile through our study sites. While



Meierbachtol. et al. (2016) found that higher order terms are not substantial at the km scale, the uncertainties related to highly localized stresses that may impact sliding friction are unquantifiable.

### 2.4.2 Basin catchment

Our analysis of basal meltwater generation is extended to the catchment scale using equation 1 along with constraints from our site-specific measurements and existing datasets for ice geometry and flow speed. As in section 2.3.1, geothermal heat flux is assumed to be 27 mW m$^{-2}$ based on nearby measurements. Because 32 boreholes drilled across the transect show temperate basal ice and wet bed conditions, we assume no thermal gradient draws heat from the bed into the ice. The Clausius-Clapeyron relation does create a very small temperature gradient which we neglect.


We approximate basal shear stress by averaging calculated driving stress over a triangular averaging window of eight ice thicknesses to roughly capture the effects of longitudinal stresses (Kamb and Echelmeyer, 1986). Driving stress is calculated based on BedMachine3 surface and bed geometries (Morlighem et al., 2017). The driving stress at this site is representative of much of western Greenland (Maier et al., 2019), and sparse direct measurements from elsewhere in western GrIS also show a high fraction

of motion results from sliding at the bed (Maier et al., 2019). Thus, we adopt the assumption that measurements showing a high sliding fraction at the 33 km site are representative of sliding across the catchment. This is an important assumption that must be acknowledged.

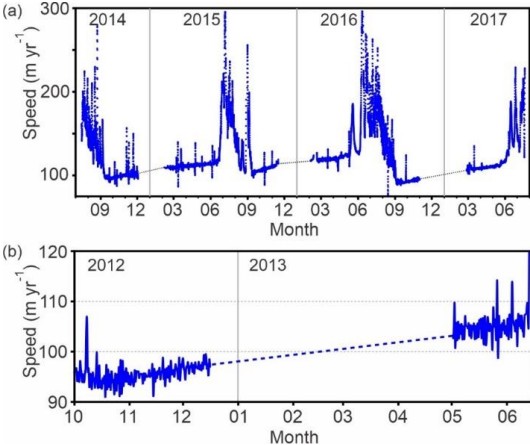

**Figure 2.** Ice flow speed during winter months measured with on-ice GPS stations and processed against base station shown in Figure 1: (a) three winters of flow speed at 33 km site; (b) one winter of flow speed at 46 km site. Dashed lines indicate missing data due to GPS shutdown due to mid-winter power loss. All measurements indicate higher speeds in late winter than early winter.

### 2.5 Cavitation and meltwater budget

To assess the fate of basal meltwater, we next we examine local storage in subglacial cavities of the meltwater generated over time. We balance the water from basal ice melt against the opening of water storage space in basal cavities. We focus this analysis on the 33 km site where observational data at the surface and in boreholes allow maximum constraint on the problem.

We assume basal sliding opens water filled cavities on the downstream side of bedrock bumps as motion over a hard bed accelerates through the winter. We implement the analysis developed by Kamb (1987) for determining cavity stability on a highly idealized orthogonal rise/step geometry. Cavity opening is a function of sliding speed and basal water pressure, and closure is due to viscous creep of the ice. The stable dimensions of water cavities depend on these processes and the geometry of the bedrock roughness.





Testing of this theoretical approach against experimental data has shown it to be a good approximation of cavity geometry (Zoet
and Iverson, 2015).

We have borehole measurements of sliding speed and water pressure to constrain the opening rate at the 33 km site, but dimensions
of the bedrock bumps hosting water cavities are unknown. However, we also have estimates of the quantity of water generated by
basal melt over time. With this added constraint, we test the plausibility that the basal hydrologic system consists of isolated water
cavities that expand to accommodate all new water generated by basal melt during the winter months. For a range of possible rise
heights, we calculate the corresponding cavity lengths at the beginning and end of the winter period, with the assumption that the
cavities are isolated and conform to the observed water pressure and sliding speeds.

The height of a water cavity is set by the step's rise, but a cavity of a particular length can potentially exist on steps of many
different lengths. Therefore, we next consider the lengths of steps in the bed's geometry for a range of possible rise-heights. The
unsubmerged portion of the step provides the sliding surface that generates basal meltwater during the winter, and the adjacent
cavity must expand to accommodate the water as it is produced. The step must be long enough to both accommodate the expanded
cavity, and also maintain an unsubmerged sliding surface required to generate meltwater as described in equation 1.

In a two-dimensional framework, the change in cavity area $\Delta C_a$ is the difference between the initial and final cavity areas during
the winter period. The cavity areas are determined by integrating the gap height of the cavity over the length of the cavity at the
start and end of the winter period, $t_o$ and $t_f$. The total meltwater, $M_{tot}$, on some step length, $L_s$, is the step length times the sum of
the water equivalent ice melt rate from frictional and geothermal heat sources, $\dot{M}$, over the time of interest such that:

$$M_{tot} = L_s \int_{t_o}^{t_f} \dot{M} dt \qquad (2)$$

where,

$$M_{tot} = \Delta C_a. \qquad (3)$$

This analysis cannot predict specific cavity geometries or the fraction of bed occupied by cavities; rather, the purpose of this
analysis is to assess the range of bed roughness values (step rise vs length) that are necessary for all calculated basal melt to be
stored locally in a closed cavity that expands over the winter in accordance with our observed water pressure and ice flow
constraints. Thus, we test the feasibility that cavities in winter are completely isolated as is often assumed.

Table 1. Characteristics of winter acceleration as measured with GPS stations.

| Year (site) | Start (dd-mm-a) | Duration (days) | Start speed (m a⁻¹) | End speed (m a⁻¹) | Acceleration (m·a⁻²) | Increase (%) |
|---|---|---|---|---|---|---|
| 2012-13, 46 km | 13-10-12 | 243 | 94 | 106 | 0.05 | 11 |
| 2014-15, 33 km | 15-09-14 | 276 | 97 | 116 | 0.07 | 16 |
| 2015-16, 33 km | 15-09-15 | 239 | 103 | 123 | 0.08 | 16 |
| 2016-17, 33 km | 06-09-16 | 261 | 91 | 113 | 0.08 | 20 |






## 3 Results

### 3.1 Winter ice flow

The winter period at the 33 km site is a distinct phase of the annual velocity cycle based on three characteristics: 1) a relatively slow average speed; 2) no diurnal or large-magnitude changes of speed; and, 3) a continuous and steady acceleration of speed (Fig. 2). The average daily acceleration during the three winters measured by GPS was about 0.07-0.08 m a$^{-2}$, persisting for 239, 261, and 276 days. A very slight decrease in acceleration was present as the winter progressed such that late winter acceleration was slightly below the winter average. The total speed increase during the winter was 11-20%, depending on the year (Table 1). The changes are well outside the GPS uncertainty (position uncertainty is 1-2 cm, yielding velocity uncertainties of ~1% over a weekly timescale). GPS measurements at the 46 km site confirmed a similar annual velocity cycle in 2012, but a slower winter acceleration with daily acceleration averaging about 0.05 m a$^{-2}$ (Table 1).

The onset and termination of the winter mode can be delineated on the basis of surface speed, surface meteorological conditions, and basal water pressure (Fig. 3). As the surface melt becomes less intense during the autumn season, the speed diminishes and eventually diurnal variations in speed cease. The speed then reaches an annual minimum, which as in Stevens et al., (2016), we designate as the transition to winter mode. This occurred on 15-September, 15-September, and 06-September during years 2014-2016, respectively. The minimum speed varied between 91-103 m a$^{-1}$, a range of 12%, depending on the year. Diurnal swings in basal water pressure end, concurrent with the end of diurnal variations in speed. However, the diurnal pressure variations are superimposed on multi-week changes that include a substantial drop from high summer values in the autumn season, reaching an annual low about 10 days prior to the speed minimum. Pressure then climbs over several weeks to high and steady winter values.

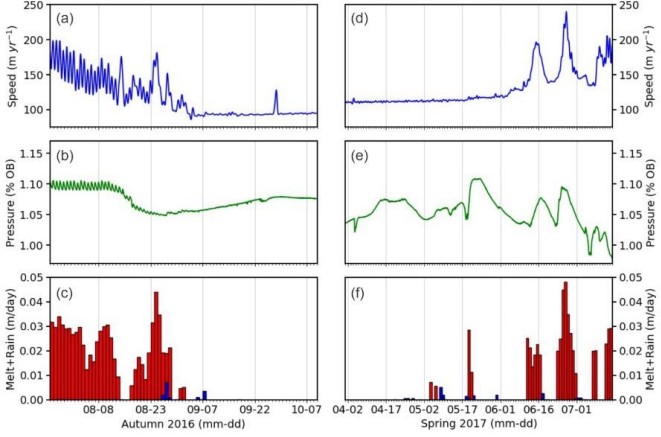

**Figure 3.** Time series of data collected at 33 km borehole site showing start and end of the winter period: (a-c) show winter onset; (d-f) show winter termination. Melt is red, rain is blue (c and f). Borehole, GPS, and meteorological instruments all located within 10 m of each other.

The winter period at the 33 km site represented 65-75% of the calendar year, accounting for about two thirds of the annual displacement. The end of winter is defined by an abrupt speed increase in spring (mid-May to mid-June) associated with the onset of surface melt. Sudden speed-up events are associated with high intensity melt periods (Fig. 3). Basal water pressure in spring becomes far more irregular than during winter, but does not consistently rise or fall and does not yet contain diurnal variations. Peak speeds in spring eventually increase threefold over winter values, approaching 300 m a$^{-1}$ (Fig. 2).

MEASuRES data reveal that ice flow acceleration over the winter of 2017-2018 occurred across a broad area of the IS basin ablation zone. The acceleration is spatially continuous across a reach from about 30 to 50 km inland from the terminus, and



extending north and south to the basin boundaries (Fig. 4). The 33 km site is near the centre of the reach, and the 46 km site is near

the furthest inland edge. Other areas of the basin may or may not have accelerated during the 2017-2018 winter: only within this reach do the magnitudes of acceleration rise above measurement uncertainty. These ice speed increases (the amounts above measurement error) are up to 10% of the winter velocity minima. The reach where winter acceleration occurs with a high degree of certainty is within the fastest moving portion of the basin, where winter speeds are >100 m a$^{-1}$.

### 3.2 Basal water pressure in winter

Basal water pressures measured directly in boreholes at the 33 km site, as well as at two other borehole sites in the IS basin, demonstrate high and relatively steady values for nearly 8 months of the winter period (Fig. 5). No consistent trends are observed in the winter pressure values. Water pressures are near or just above overburden and undergo very low frequency variability over periods of weeks to months. The small variations likely stem from the advection of pressure sensors 10s of meters across variable

bed conditions. The character of winter pressure variability strongly contrast with the summer period, which has large and frequent changes in pressure and includes diurnal swings (Wright et al., 2016).

### 3.3 Basal meltwater production

The rate of basal meltwater generation at the 33 km site increases all winter, commensurate with the steadily increasing ice sliding

speed. Whereas the geothermal heat flux for the region is estimated to be ~27 mW m$^{-2}$ (Meierbachtol et al., 2015) the sliding friction during winter is many fold greater, calculated to be about 340 mW m$^{-2}$. Consequently, the time series of meltwater generation largely mimics the ice speed series.

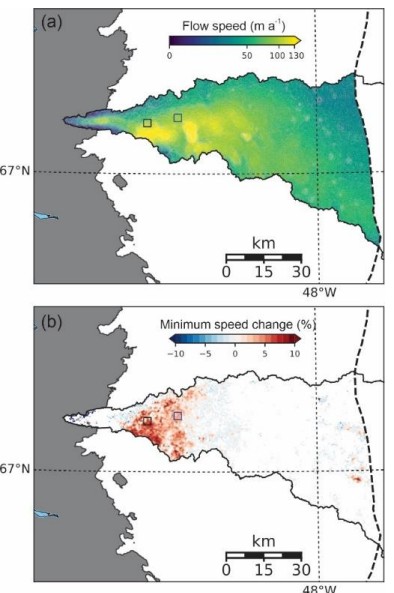

**Figure 4.** Spatial characteristics of ice flow across Isunnguata Sermia basin during winter of 2017-2018 based on MEaSUREs v2 product. (a) pattern of ice flow speed during early winter measured over month of October 2017. (b) minimum ice speed change (as percentage) from October through April; shown are the percentage speed changes minus the potential percentage change resulting from error terms. Areas showing no change could have potentially experienced a October-April speed change, but not enough to overcome errors. The 33 km and 46 km measurements sites are shown as boxes.

The basal meltwater generation during the 2016-2017 winter, the slowest of the three measured winter periods (Figure 2), increases from 2.8 cm a$^{-1}$ at the start to more than 3.5 cm a$^{-1}$ by the end of winter (Fig. 6a), a 20% increase over the period. The cumulative meltwater for the winter months reached 2.6, 2.4, and 2.3 cm during winters of 2014-15, 2015-16, and 2016-17, respectively. Meltwater generation differs from year-to-year, stemming from the date of the winter onset and the initial starting speed. The



instantaneous ice melt rate on any particular day of winter can vary by >10% between years. However, the cumulative meltwater
at any point in the winter is not particularly different between years since the winter is not long enough to cause much spread from
the rate differences (Fig. 6b). Note that all above values represent an area-average. In reality, some area of the ice is separated from
the bed over basal water cavities, meaning that the thickness melted from ice remaining in contact with the bed is greater than this
average. Furthermore, rates of basal ice melt can be expected to be up to three times greater during high-speed periods of summer.

The highest rates of calculated basal friction and basal meltwater generation in the IS basin are concentrated around the 33 km site,
the region experiencing the greatest acceleration in winter. Here, the flow speeds are relatively high and the basal shear stress is
large due to comparatively steep surface slopes, and so basal frictional heat is higher than elsewhere in the basin (Fig. 7a). Basal
frictional heating in this region is >600 mW m$^{-2}$ over areas that are 10s of km$^2$. Frictional heating is far lower further inland, but
still several times greater than geothermal heat flux. The values calculated across the basin are likely upper limits, because the
fraction of motion attributable to sliding is not likely any higher than what we measured at the 33 km site.

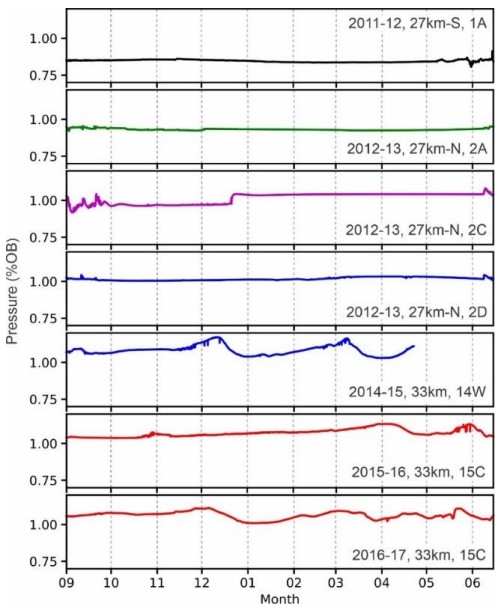

**Figure 5.** Basal water pressure during winter months, measured in boreholes. Year, location as distance (km) from the terminus, and borehole name shown on right side of each panel.

The highest calculated bed friction across the basin is sufficient to generate >5 cm of basal meltwater during the October-April
winter period (Fig. 7b). The frictional heating from slower speeds further inland than about 75 km generates lower melt amounts
of 1-2 cm per winter, but this part of the basin has far greater area than closer to the margin. Averaged over October-April, the
estimated melt integrated over the entire basin is equivalent to an instantaneous meltwater flux of 3.9 m$^3$ s$^{-1}$. The total volume of
water generated from the IS basin over 2017-2018 winter is calculated as 5.2e7 m$^3$. While the basal meltwater generation in some
areas of the basin can increase over winter by 20% (i.e., the accelerating region surrounding the 33 km site), the total meltwater
from the basin mainly reflects the large inland area of the basin experiencing little to no winter acceleration (Fig. 4). Thus, neither
the instantaneous rates or winter totals should vary much from year-to-year.

The potential demand on a subglacial drainage system for exhausting meltwater generation is demonstrated by the accumulation
of the melt rate from top-to-bottom of the basin (Fig. 7c). Thus, if all basal melt is routed to the terminus, the accumulated values





are the required flux of the drainage system with respect to position along the basin. Notably, values reach 3.0 m³ s⁻¹ by >50 km inland from the margin and do not increase much over the last 25 km because the basin width narrows toward the terminus. Note that we have not accounted for any meltwater generated above the ELA, where sliding speeds are relatively slow but the bed is most likely at the melting point to an eventual boundary with frozen conditions (MacGregor et al., 2016). Furthermore, we have accumulated all basal meltwater, whereas some water may be routed into the groundwater system, or stored locally in the ice or in the drainage system. We discuss these issues in section 4.1.


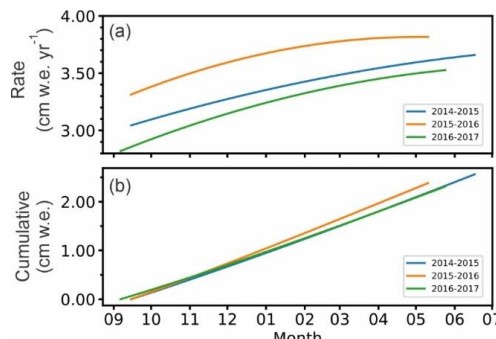

**Figure 6.** Basal meltwater generation at 33km site: (a) instantaneous basal melt rate in water equivalent over the winter during three different years; (b) Cumulative meltwater generation over the winter period during three years.

### 3.4 Cavitation with water budget

Functional relationships between various geometric parameters of the stepped cavity system can be derived from our observational constraints with the assumption that all generated melt is stored locally. Higher rises on bed steps generate basal water cavities that
span longer distances across the bed, with cavities that are always extraordinarily long relative to their height (Fig. 8b). For example, a 1 m rise requires cavities that extend 32-34 m across the step. This predominantly reflects the high sensitivity of cavity geometry to the pervasive high basal water pressure we measure.

With cavities submerging a portion of the bed, the remaining portion of the step in contact with ice must accommodate the sliding
friction. Because our assumption requires a closed water budget, cavity expansion constrains this length: the water generated by ice melt on the bed surface must balance the water entering storage as cavities expand. Using our site-specific calculated basal melt as constraint, we calculate the step length required to yield the basal melt that fills space opened by cavity expansion during winter acceleration (Fig. 8c). Steps any longer than this would yield more meltwater than the available cavity space. The calculated results for three years with different speeds illustrate the sensitivity of the analysis to variations in sliding speed and winter
acceleration.

Finally, we compare the length of cavities to the length of steps to limit the solution space to plausible geometric scenarios (albeit, with the no flux assumption). The ratio of cavity length to step length must be less than one because no cavity can be longer than the step it occupies. Depending on the year, the minimum step rise to produce cavities that can grow to sequester all the water
produced by basal meltwater is 0.45 to 0.7 m (Fig. 8d). However, the lengths of cavities corresponding to reasonable steps are extremely long: for a 1 m step rise, the step length is 50-75 m and the cavity length is 33-34 m.





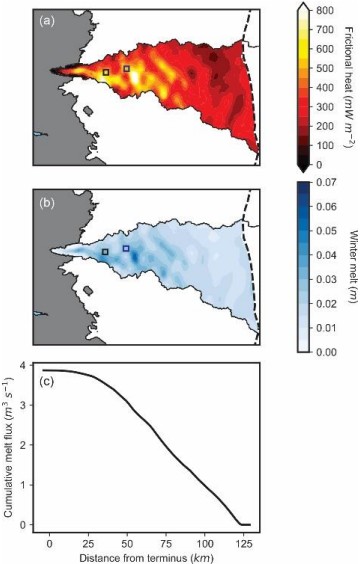

**Figure 7.** Basal ice melt during winter of 2017-2018 across the Isunnguata Sermia basin based on satellite velocity and model assumptions described in text: (a) Basal frictional heat averaged over the winter period (October through April); (b) total basal meltwater generated during the winter months; (c) Top-to-bottom accumulation of basal meltwater generation rate averaged over the winter period. Basal meltwater is integrated as a function of inland distance, irrespective of potential routing pathways.

## 4 Discussion

### 4.1 Accommodation of basal meltwater

Friction-derived meltwater is manyfold greater than the component of meltwater derived from geothermal heating due to the high fraction of motion from sliding in winter. Moreover, the rate of meltwater generation at borehole sites across the central region of the IS basin can increase by up to 20% over the winter. The winter state of the basal drainage system must therefore accommodate and/or evacuate a water flux that is likely to be reasonably large when integrated over the basin, and increases over the ~7-month period.

Some meltwater is likely accommodated by the underlying groundwater system, although groundwater modelling of fractured bedrock conditions beneath the GrIS and similar ice sheet settings (Jaquet et al., 2019; Lemieux et al., 2008) suggest recharge rates of mm a$^{-1}$ rather than the cm a$^{-1}$ rates of meltwater production. Englacial storage space is also unlikely to have sufficient capacity for the water. To accommodate the water in the overlying ice mass, englacial storage space would need to open over time to match the continuous production of meltwater. However, the stored water would need to periodically drain to avoid continuous build-up of water so as to maintain the liquid water content in the temperate ice layer below a few percent (Brown et al., 2017). Further, our borehole transect reveals the temperate layer at inland locations can be just a few meters thick (Harrington et al., 2015), which severely limits the potential for liquid water storage in basal ice (Downs et al., 2018).

The total water in storage and the geometry of cavities is a function of unknown bed roughness. Nevertheless, an increase in the storage space over the winter should occur due to cavity expansion as the ice flow accelerates (Iken, 1981). Our measured basal water pressures are continuously near overburden and sliding speeds are relatively high. Thus, Kamb's (1987) stepped-cavity system with a no-flow condition requires that stable cavities have large step heights/lengths (e.g., cavities many 10s of meters long). Only these large sized cavities can undergo sufficient expansion to accommodate the volume of basal meltwater generated. The large size, however, is unsupported by prior observations of cavities on a hard bed (e.g., Anderson et al., 1982; Walder & Hallet, 1979) which suggest water cavities are much smaller. Observations of a single cavity at our 33 km site using dye dilution techniques also indicate a far smaller cavity size with a volume of several m$^3$ (Meierbachtol et al., 2016). Furthermore, the large





cavity size is radically inconsistent with bed geometries considered by subglacial hydrology models (e.g., Downs et al., 2018;
Schoof, 2010; Werder et al., 2013).

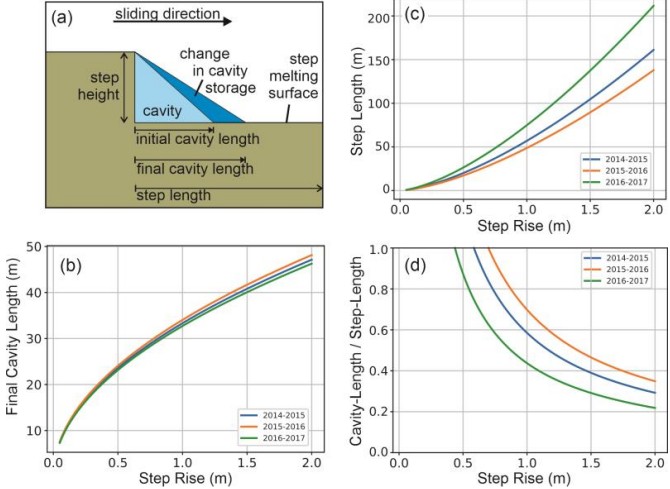

**Figure 8.** Cavity geometries based the Kamb (1987) stepped-bed analysis. Shown are the stable cavity lengths required to match sliding speed and basal meltwater generation in three different years under the assumption of isolated cavities (a). Results are given for a range of possible step heights. (b) final cavity length after cavity has expanded to accommodate the meltwater generated over winter period; (c) length of steps required to accommodate both the water cavity and the portion of the step in contact with ice; (d) ratio of cavity length to step length, which must be less than 1 because cavities cannot be longer than their steps.

A more realistic irregular bed would yield shorter cavities than the over-simplified stepped bed, because the downstream side of cavities would terminate against any slight rise in the bed. Accommodation of new water though cavity growth would be even
more restricted and so the Kamb (1987) stepped bed analysis represents upper-end of potential cavity storage of the water generated by basal melting. Hence, the scenario that all basal meltwater is accommodated by opening isolated cavities appears untenable. The challenges of attributing all winter meltwater generation to local sinks in ice, substrate, or expansion of water cavities, therefore implies connectivity in the subglacial drainage system during winter.

Airborne radar reflectivity suggests widespread basal water storage during the winter along the non-trough areas of the IS basin (Chu et al., 2016). Indeed, water emerging from the IS basin outlet has been observed in winter, although it was attributed to release of stored water with no consideration of water flux due to basal ice melt (Pitcher et al., 2020). Minor meltwater fluxes should be routed locally through an orifice-connected cavity network (Kamb, 1987). Basal melt rates are sufficiently high that the water accumulating in hydraulic potential lows as it migrates toward the margin could reach discharges on the order m$^3$ s$^{-1}$ (Fig. 7). For
example, if all basal meltwater from the IS basin were routed to the terminus outlet, the mean winter discharge at the outlet would be about >3 m$^3$ s$^{-1}$. Discharge of this magnitude, if focused into a single or small number of pathways, (c.f., connected cavity network), has potential to support some degree of persistent channelization (Hewitt, 2011; Meierbachtol et al., 2013). Thus, overwintering of one or more basal meltwater channels in favoured locations is quite plausible, although flux limitations demand they are rare.



### 4.2. Implications for acceleration in winter

Beyond our observations of winter speed up, a steady increase in speed over the course of the winter has also been reported for certain locations of the western Greenland ice sheet with satellite (e.g., Fitzpatrick et al., 2013; Joughin et al., 2008, 2010; Moon et al., 2014) and ground-based (Stevens et al., 2016; van de Wal et al., 2015) observations. Acceleration of ice flow during winter is not an omnipresent characteristic of glaciers, but neither is it unique to Greenland: individual reports of speed increases throughout winter also exist for surge type glaciers in North America (Abe and Furuya, 2015) and for valley glaciers in the Alps (Iken and Truffer, 1997). While the phenomenon is known to exist in Greenland, the driving processes are unclear, particularly with regards to in situ observations of bed conditions.

Ice sheet acceleration can result from an increase in driving stress and/or decrease in basal traction. We see no way to link our observations of winter acceleration to a steady increase in driving stress, because the speedup is widespread and irrespective of ice thickness gradients. Earlier works have appealed to pressurization of the basal drainage system over the course of winter as a means for decreasing basal traction, manifested conceptually through either isolation of cavities as the drainage system shuts down (e.g., Fitzpatrick et al., 2013), or widespread re-pressurization of the bed in response to closure of low pressure channels (e.g., Sole et al., 2013). Our observations suggest pressurization occurs over a few weeks in autumn (Figs. 3, 5), but then no monotonic increase in the basal water pressure over the winter period (Fig. 5), a finding in agreement with observations from elsewhere in Greenland (Ryser et al., 2014).

Ice acceleration during winter, however, implies cavity expansion (Iken, 1981) and the transfer of load to unsubmerged bedrock, albeit on a cavity network with some connected water flux. This leads to the question of whether a steady cavity opening resulting from basal meltwater generation drives the steady winter acceleration of ice flow. Water evacuation through a poorly connected drainage system would modulate the acceleration and maintain the constant high pressure. Such processes should be annually repeatable, but will vary from place-to-place due to basal sediment cover and roughness. Indeed, the strongly accelerating region of the basin has a hard bed where we have drilled boreholes (Harper et al., 2017), whereas the bed is sediment covered both further inland (Booth et al., 2012; Kulessa et al., 2017) and closer to the margin (Dow et al., 2013).

### 5 Conclusions

Surface and borehole measurements at an instrumented site in the Isunnguata Sermia basin constrain estimates of basal meltwater generation and subglacial hydrological conditions over the winter period. Despite relatively slow surface speeds during winter, basal ice melt from sliding friction is many times that resulting from geothermal heat flux. Heat is available to generate ~2.5 cm of meltwater during the winter period, but amounts differ from year-to-year and are linked to the date of the winter onset and variable ice flow speed at the start of winter. We observe a slow and steady acceleration during winter, similar to reports from elsewhere on the Greenland ice sheet. As a consequence, the instantaneous melt rate on any given day of winter can vary by >10% between years. Further, an acceleration of ice flow over the winter can increase the instantaneous melt rate by 20% from the start-to-end of winter.

Borehole measurements show steady and high basal water pressure during winter. Ice and groundwater sinks do not likely have sufficient capacity to accommodate the basal meltwater generated during winter where rates are high. Cavity opening associated with ice acceleration, accommodates some meltwater, but a residual is likely routed through a connected drainage system and



discharged to the terminus. The winter acceleration is thus associated with opening of water cavities, but in a connected and high-pressure system. This state of the winter drainage system sets the initial conditions for the onset of spring drainage evolution and speed up.

Variations in driving stress and winter speed across the basin drive a spatial pattern of basal meltwater generation. Assuming the
sliding rates we measure at our field site persist over the basin, greatest melt rates are concentrated ~30 to 50 km inland from the terminus during the 2017-2018 winter; a region that also experienced a measurable acceleration during winter. The water generated from the IS basin over the winter is calculated to be 5.2e7 m$^3$, or 3.9 m$^3$ s$^{-1}$ averaged over the winter. Despite our measurements of significant time variability across a broad reach of the basin, the basin totals mainly reflect the large area experiencing comparatively low basal melt rates and having little to no winter acceleration. Thus, if all meltwater from basal ice melt across the
basin is routed to the terminus outlet, we expect the total discharge there due to basal ice melt does not increase much over winter or vary much from year-to-year. Substantial time variability, if present, would likely reflect release of stored water.

*Code/Data Availability*: Data used to generate this manuscript are available for download from the Arctic Data Science Center
(https://arcticdata.io): borehole water pressures (doi:10.18739/A2VQ2SB3C); GPS surface velocity data (doi:10.18739/A2154DP90); and, meteorological data (doi:10.18739/A2QV3C418).

*Author Contribution*: JH and TM conceived of the work. NH, JS and AS assisted with data collection and analysis. All authors contributed to paper preparation and refinements.

*Competing Interests*: The authors declare that they have no conflict of interest.

*Acknowledgments*: This project was funded by the NSF Office of Polar Programs-Arctic Natural Sciences awards #1203451 and #0909495, Svensk Kärnbränslehantering AB, and Canadian Nuclear Waste Management Organization.

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
