# Peer review of "Generation and Fate of Basal Meltwater During Winter, Western Greenland Ice Sheet"

_The Cryosphere, 2021_

## Referee Comment (RC2)

[referee-annotated manuscript omitted]

---

## Author Comment (AC1)

**Comment on tc-2021-179**
**Martin Lüthi (Referee)**

Referee comment on "Variability of Basal Meltwater Generation During Winter, Western Greenland Ice Sheet" by Joel Harper et al., The Cryosphere Discuss., https://doi.org/10.5194/tc-2021-179-RC1, 2021

General Comments

This is an interesting attempt to investigate, from in-situ measurements, the basal meltwater generation under the Greenland ice sheet during winter. While there is room for improvement, I think the manuscript can be published after taking into account the comments outlined below.

In Equation (1) the melt rate seems to be volumetric. This should be explicitly stated (units for all quantities). Also state that this is the conservation of energy in a small volume around the glacier bed.

The development in Equations (2) and (3) needs to be improved. Units for the quantities should be given (is M_tot a volume, ice equivalent or water equivalent, or mass?). As it is, it seems that volume (or melted mass) and area are confounded in (3), and there are tacid assumption on ice thickness. Also, it is not clear why all the subscripts are needed. A small sketch of the geometry would help clarify the situation. Figure 8a shows the envisioned geometry, but does not help in that respect since the quantities are labeled with words, without relation to the symbols used in the equations.

Also, the formulas (2) and (3) seem to be unrelated to the results. It seems that some kind of force balance is needed to properly calculate melt rates and cavity size, as larger cavities lead to larger stresses outside the cavities, and incresed melt rates there.

The whole process is also likely spatially heterogeneous due to longitudinal and lateral stress transfer (e.g. Ryser et al, 2014 a,b). This might somehow be implied in the equations, but should be made explicit.

We obviously provided inadequate information to the reader regarding this aspect of the paper (we apologize to reviewers for this tactical error). While we follow the published cavity analysis of Kamb (1987), it was not reasonable on our part to assume the reader would be familiar with details of that analysis and how we implement it. We have taken three steps to address this shortcoming of our manuscript:
>    1. We added text to make clear the dimensions of various terms (which do balance).
>    2. We added more text and equations describing Kamb's cavity model and our implementation is better described with equations and not just words.

3. We replaced figure 8a with a new figure to demonstrate cavity analysis approach, define of the various terms, and show an actual modeled cavity.

One further point to take into account: The melt will be generated not just at the bed (as assumed in Eq. 1), but also within the ice column, especially where vertical shear rates are high. I know that at the drill site this component is very small, but this might not be the case on upstream or downstream obstacles. As a reference quantity, to which these results could be compared to, it would be interesting to caluculate total dissipation of potential energy. This is proportional to u_avg * H * slope. This quantity would also give a good estimate on the percentage of heat generated at the bed with respect to total heat dissipated.

Indeed, strain heating in the ice column can be a large heat source – certainly in locations with high vertical shear and also in places with strong horizontal or lateral shear. However, the majority of the ice column is quite cold (i.e., -18C) and so the heat generated in such locations is used to warm the ice. What matters here is the strain heating in the fraction of the temperate layer with hydraulic connectively with the bed. The connectivity/storage can exist as grain-scale voids (i.e., connected three grain intersections) and/or macro-scale voids (i.e., basal crevasses or perhaps 'vugs' which are sometimes observed in emergent ice in ablation zones. Little is known about either of these englacial storage mechanisms. As we state in the paper, however, even if we consider then entire temperate layer, we have the problem that the layer is only a few meters thick and new storage space must for some reason open over time as meltwater is generated, and then seasonally drain to avoid build up.

We understand the question here because englacial water storage/connectivity remains a poorly understood aspect of glaciology. And, Figure 7a would seem to be repetitive with the suggested calculation. Thus we feel that if we were to attempt to expand the discussion of these issues beyond the paragraph we already have, we would divert the reader's attention sideways without making more progress.

Specific Comments

40 This was estimated to be an order of magnitude higher in Lüthi et al (2003).
We added this fact and reference.

46 What are storage sinks within the ice? Basal crevasses?
We added a specific example, "e.g., basal crevasses", but left the sentence structured as generic regarding storage. This is because many ice sheet models have a term for generic water storage in ice, often conceptualized as grain-scale storage, and is sometimes set at up to 10%!

Figure 1, Caption: are all used data sets indicated here? Ordering them according to data set would make this clearer (e.g. water pressure at 27N, 27S and 33; GPS at 33 and 46).
Caption reworded for clarity.

Figure 1: dark blue on dark blue (station 46) is hardly visible.

We changed the darkness of the station box to provide more contrast with the background.

113: The term "fitted" is better expressed as "instrumented with" for an international readership.
Appreciate this perspective – changed.

116: Why not just use pressure? Then it is absolutely clear what you are comparing.
Prior publications have presented borehole water pressure measurements in units of absolute pressure, effective pressure, meters of water equivalent (above bed or below surface), and percentage of overburden. Each unit has its pros and cons. Indeed absolute pressure does provide a direct comparison between sites at a uniform scale, but assessment of the numbers is challenged by the fact that the pressure is closely related to ice thickness which is highly variable between locations. Thus, we believe that here the most appropriate presentation of pressures concerning the focus of the paper is to scale the pressures overburden. Had we been testing sliding laws, effective pressure would be better). Further, this unit is consistent with our prior publications for this region such that the new pressures presented here may be more easily compared.

125: What are the units of \dot{M}?
We have added this specific information, and have expanded our overall presentation of this section.

142: You could add some qualifying statements that with longitudinal and lateral stress transfer could change local shear stress by a large percentage below/above average values. But on the km-scale this is likely to be averaged out. But then it is not clear, what the borehole deformation measurements mean.
We have added this wording to the next paragraph (starting at line 147 rather than 142).

150: also Ryser (2014a)
Reference added.

151: In light of the above comment, this need not be true, and internal deformation is likely exceptionally low at the drill site (as compared to e.g. Ryser, 2013). It would be interesting doing the same calculations with their results for comparison.
We have added the reference (Ryser, 2014 because we could not locate a Ryser, 2013), and words to qualify our statements.

156: there is a stray "we"
Fixed.

184: This seems unintuitive. Melt only happens over \lambda - L_s,
\lambda being the wave length, and there is no melt over a cavity as there is no friction. Then also Equation (3) is unclear. This is only true if there if the water in the cavity has unit thickness. M_tot is a volume, and C_a is an area, so units do not match.

This entire section has been redone to address our lack of clarity.

Table 1: It would be useful (and more robust) to show total displacement during the winter period. This displacement times frictional stress gives the total energy released at the base, which is readily transformed to total melt during that period.
We are reluctant to divert the reader's focus from the characteristics of the acceleration during winter, which is also an important aspect of the paper. However, we understand your point and so we have added text to results section concerning displacements.

240 more precisely: "the heat generated by sliding friction"
Changed as suggested.

Figure 4b: squares are barey visible, consider using a better color
We increased the line weight three fold. The squares are now much more visible.

267 (also 397): typeset the number properly, not in computer coding
Fixed.

292: how are these values calculated. Give complete formulas that allow the reader to repeat the calculations. Why is no reference made to Eqs (2) and (3), where this whole development should be carefully made.
We agree that our presentation was insufficient. To remedy this, we have expanded our description of the methodology as described above. This paragraph now includes call out to the appropriate equation, and edited wording which hopefully improves the clarity.

297: step length should be named "wave length" as some kind of periodic bed is assumed.
We agree with the logic behind this suggestion. However, since we apply the analysis of Kamb (1987), we believe it is best to follow the terminology used in that paper, which is step/rise, length/height, etc. To adopt different terminology could confuse a reader who wishes to cross reference our work with the Kamb paper. Further, we note that Zoet and Iverson (2016), who also apply the Kamb (1987) model to their own work, also follow the Kamb terminology so it seems that a precedence has been set.

Figure 7, caption: This is not "heat", but "heating rate", "heat production rate", or similar.
Fixed.

310: it seems that 2 cm of water are not hard to store locally in sediment-filled basins or subglacial lakes in depressions. As water pressure is very high everywhere, this is a possibility that should be considered.
We have expanded the last paragraph to accommodate this observation (and the related comment regarding line 325).

317: Here seems to be a confusion between water stored within the ice matrix (Brown) and water stored in discrete cracks, e.g. basal crevasses of some sort or other, sponge-like and large, void space within the compact ice.
Brown et al.'s (2017) radar methodology for determining liquid water content in the ice is incapable of distinguishing the scale of the water, except for very large and separated basal crevasses yielding isolated reflectors. Thus, our aim here is to focus the reader on the amount of water while remaining appropriately ambiguous about the characteristics of the storage locations.

325: There are large depressions (100s of meters across) that could easily accommodate large volumes of water.
We have expanded the last paragraph to accommodate this observation (and the related comment regarding line 310).

330: But these might be unrealistic. Looking at the proglacial terrain shows very irregular bed undulations with very large and deep valley, deep and steep valleys etc. This looks very different from Kamb, Iken etc theories, and also from all these model inputs. Such bedrock topographies can, by moderate changes of water pressure, lift the ice and easily accommodate very large volumes of water.
We think it is important to discuss our findings with regards to common assumptions of subglacial hydrology models. We do, however, offer several qualifications to our discussion that address our point above: total water storage and geometry of subglacial water cavities is a function of unknown bed roughness; rounded sinusoids would store less water than our idealized cavity network, making our results an upper limit; and, some water could be stored in larger depressions.

354: "whereas they are small or absent at other locations (Ryser, 2014a).
Evidence and reference added to text.

365: Or lower effective pressure in subglacial sediments leading to liquefaction. A constant pressure and diffuse within the sediments, and increase the vertical extent of "softer" sediments. Such a process would readily explain the observed acceleration.
Our figure 5 and the data in Ryser (2014a) do indicate a decrease in effective pressure, commensurate with the observed sliding acceleration. Agreed that sediment deformation could be a key aspect of winter flow; we state this in the next paragraph.

373: This strongly depends on the time scales of cavity formation and cavity closure. It would be instructive to calculate approximate values to understand their magnitude and their changes under changing pressures and sliding speeds over winter.
Yes, the Kamb (1987) analysis includes cavity opening from sliding and creep closure of cavities, and we have piped the accelerating sliding speed over winter into these calculations (Figure 8)

Figure 8: step length could also be named wave length of the bed
See justification for naming this 'step' above, under line '297'.

---

## Author Comment (AC2)

Comment on tc-2021-179
Samuel Doyle (Referee)

Referee comment on "Variability of Basal Meltwater Generation During Winter, Western Greenland Ice Sheet" by Joel Harper et al., The Cryosphere Discuss., https://doi.org/10.5194/tc-2021-179-RC2, 2021

Review of Harper et al. "Variability of basal meltwater generation during winter, Western Greenland Ice Sheet.

This manuscript by Harper et al. brings together previously-published datasets with new data and analysis to investigate whether subglacial cavity expansion caused by basal melt can explain the often-observed – but never satisfactorily explained – winter-time acceleration of surface ice velocity in Greenland. It makes an original and important contribution to solving this problem. The hypothesis that subglacial cavities become hydraulically isolated during winter is tested and the authors suggest that some interconnections must remain as melt volumes exceed plausible cavity storage volumes. Important points are made regarding basal melt variability, particularly that the majority of total basal melt is generated in areas with low and invariable basal melt rates, and that this will result in relatively constant winter-time discharge fluxes at the terminus. It follows that substantial perturbations in basal melt must represent the sudden release of subglacially stored water (e.g. drainage of subglacial lakes). The manuscript is concise, well written, well-presented and the arguments are framed well within the inherent limitations of the dataset. I have a number of comments detailed below.

**General Comments**

▪ Further description of the GPS filtering and basal shear stress calculation are required to ensure reproducibility.
We have added details concerning the binning and smoothing of GPS data used to produce our velocity curves. And, we have added an equation and defined all of its terms to better describe the calculation of basal shear stress.

▪ Expansion on the application of the Kamb (1987) analysis for determining cavity size could be helpful, for example by giving some of the original equations to explain the basis of the analysis. At present the description is limited to just two equations (Eq. 2 and 3).
We obviously provided inadequate information to the reader regarding this aspect of the paper (we apologize to reviewers for this tactical mistake). While we follow the published cavity analysis of Kamb (1987), it was not reasonable on our part to assume the reader would be familiar with details of that analysis and how we implement it. We have taken three steps to address this shortcoming of our manuscript:
      1. We added text to make clear the dimensions of various terms.
      2. We added more text and equations describing Kamb's cavity model and our implementation is better described with equations and not just words.

3. We replaced figure 8a with a new figure to demonstrate cavity analysis approach, define of the various terms, and show an actual modeled cavity.

▪ Previous studies which reported and attempted to explain the winter-time acceleration in ice flow should be introduced in the Introduction – this would highlight to the reader the research gap that this investigation fills.

We have substantially expanded the opening paragraph of the introduction to address this comment. We provide references for the observation of winter speed up on both glaciers and Greenland ice sheets. Some of this text was moved from the discussion since it is more appropriate here.

▪ The Introduction (and discussion) could be expanded slightly to introduce current theory and modelling relevant to the behaviour of subglacial cavities during winter to set the scene for this study, and to show how this study contributes to our understanding.

We understand the rationale behind this comment and we deliberated on this issue for some time. In the end, we are hesitant to send the reader in too many directions in the introduction: topics that include basal ice melt, winter ice flow acceleration, and also subglacial hydraulics. Our paper does not present new advancements on understanding of subglacial hydraulics and the cavity analysis we perform is nothing new; it was developed by Kamb (1987) and subsequently applied by other workers (e.g., Zoet and Iverson, 2015). Thus, the discussion presents some implications with references of our findings, but does not delve deep into a full overview of subglacial hydraulics. The third paragraph of the introduction is devoted to our paper's contribution in this area -- an assessment of the potential sources and sinks for basal water during winter.

**Specific Comments**

Please find minor typographical corrections and suggestions in the marked up PDF attached.
→ We have addressed all of the copy editing remarks included in the supplemental PDF. We are very appreciative of these comments and the time invested in careful reading and commenting on the manuscript.

L1 – In the title "western" should be lowercase as there is no "Western Greenland Ice Sheet". Also, this paper does more than just measure basal melt variability and its main take home point relates to cavity dynamics and subglacial hydrology – the title could be revised to reflect this better.

We have made sure that "western" is lower case in the manuscript body (but the first letter of each word in the title is capitalized). We have edited the title to reflect the paper's broader scope and implications.

L15 - Can you mention in the abstract that you calculate cavity dimensions/dynamics and compare them to melt volumes to determine whether or not all basal meltwater can be accommodated by cavity growth. At present you just refer to "insights into subglacial hydrology": I think you can be more specific. You might also mention in the abstract or introduction that you specifically test the hypothesis that cavities remain hydraulically isolated in the winter.

We do already mention that we perform 'analysis of basal cavity dynamics'. Our issue with adding more is that the manuscript guidelines say that abstracts should be 150-250 words, and we are already at 249 words.

L36 – you should introduce here previous observations of, and explanations for, winter- time acceleration in ice velocity. You currently do this on L352-358 but it should come earlier as it is key to the data and analysis presented. The winter-time acceleration is evident in most measurements of ice velocity (e.g. Sole et al., 2013). Note that there are also detailed winter velocities from near your study site but 140 km inland that show winter acceleration presented in Doyle et al. (2014). Winter acceleration in ice velocity is also presented in Phillips et al. (2013; Fig. 4), which I believe is mis-interpreted therein as being caused by increased deformation due to heating of the ice due to warming ice temperatures. The first paragraph of the introduction could introduce the literature on this topic slightly better, which would set the reader up for the analysis to come.

We have made this change to the manuscript: we moved the text introducing the winter acceleration (L353-358) from the discussion to the introduction where it is more appropriate, and we expanded the wording.

With regards to Phillips et al. (2013), Fig. 4 supports our findings but we believe it presents a completely aliased representation of winter ice flow, since it has just four data points spanning three different years. Sole et al. (2013) focus on mean winter velocity, but do not discuss winter accelerations. Looking at their Fig. 2, their site 4 does indeed seem to demonstrate a steady increase in speed all winter like we observe, whereas all other sites show none. Thus, our Fig. 4 is used to examine the variability around the basin. The winter accelerations in van de Wal et al., (2015) are circled in a figure showing speeds, but never discussed in a caption or the text. These examples demonstrate the challenge of trying to summarize all that's been measured and presented about winter acceleration, which we must do briefly since our paper in mainly about the fate of basal melt. Thus, we think it is best to simply provide some general summary statements and example references as we have done. The community needs a full review paper about ice flow acceleration in winter because there are so many different scraps of observations scattered about.

L69 – delete "full". All years have winter gaps due to power outage.
one

L73 – what is meant by "bed framework"?
We have changed this to 'composition of the bed-surface'

Figure 1 caption and elsewhere – consistency with "basal water pressure" and "borehole water pressure" would help the reader who doesn't know these are assumed to be the same thing.
We have eliminated use of 'borehole water pressure' in favor of 'basal water pressure', with section 2.3 describing how basal water pressures were measured in boreholes.

L93 – The methods used to filter and differentiate GPS position data need to be detailed to allow the study to be reproduced.
We have added this and additional information about the GPS processing.

L124 – why not rearrange to get M on its own on the LHS?
The equation is expressed as the energy balance at the bed, and we have added wording to make this clear. We note that this a common way to express the relationship (e.g. Binschadler et al., 2011, J. Glac.; Paterson, Physics of Glaciers).

L126 – More detail on how the basal shear stress was calculated (e.g. the equation used) needs to be given.
We have added an equation and defined all of its terms to better describe this calculation.

L128 – there is avoidable repetition here with L142-144 and neither spell out why basal temperate ice presents a barrier to upwards heat conduction, which is due to the Clausius-Clapeyron gradient causing a reverse (and small) temperature gradient.
We have edited this text to avoid repetition.

L185 – add an example reference to support the statement that cavities are often assumed to become isolated during winter. In general, slightly expanding the discussion of the theoretical understanding and modelling treatment of cavities and basal melt in winter would boost the significance of this papers' findings.
This section has been redone to provide more detail on the cavity analysis methods, with more explanation and additional equations. We provide the rational for the cavity analysis as a test of the end-member closed water budget. Also, see response to bullet four above.

Table 1 – "Site" needs to be taken out of brackets and put into a new column.
Done.

L199 – I'm not sure you mean "daily acceleration", will "acceleration" suffice?
Yes, we have changed the text as suggested.

L209 – The seasons have strict definitions in meteorology, and it would be helpful to note early on that the terms "spring" and "winter" are being used more loosely than their normal strict definitions, that is to reflect the period at melt-onset and the period when the subglacial hydrology is in its "winter-mode".

This information has been added to the second paragraph of the methods.

L213 – Move descriptions of water pressure results to the section on water pressure (or remove subsections altogether). It would be good to expand on the description of winter- time pressure variability due to its relevance for cavity opening.
We have moved this text as suggested, and edited the section on basal water pressure for clarity.

L241, L255, L263 – "sliding friction", "basal friction", and "bed friction" should be "frictional heat from sliding".
These have all been changed as suggested.

L269 – observations of inter-annual variability in winter-time acceleration in ice flow at 140 km inland are presented in Doyle et al. (2014, Fig. 2). Can these measurements help quantify what is meant by "little to no winter acceleration"?
These data would be a nice example to present to the reader concerning our point. However, the S10 site located about the ELA and therefore beyond our study domain.

L323 – With the uninformed reader in mind, what are sliding speeds high relative to? Perhaps cite your previous work on this.
We have added details to make this clearer.

L328 – Note that cavities accessed (or even created) via drilling may not be representative of the majority of natural cavities. Analysis of the glacial foreground or subglacial topography in West Greenland would provide a better picture on the basal roughness; such analysis has been done (e.g. Lindback, 2015).
Agreed that drilling can create an artificial condition. The paper we reference discusses this and argues that while no subglacial measurement can be considered undisturbed, at least we know that the borehole was advected 10s of meters from the original drill site. We offer this fact as essentially an anecdote – the one attempt we know of to measure the size of a subglacial cavity in Greenland. We do not hang our results or interpretation on this single, and perhaps flawed, observation. The Lindbäck paper discusses roughness at larger scale than individual cavities. However, in response to comments by reviewer 1, we have added text two paragraphs later to address the issue of basal roughness.

L367 – you could spell out for the reader even more clearly that previous explanations are not sufficient to explain all the observations.
This is a good point and we have added explanatory text suggested.

L374/375 – mention that the studies reporting sediment are also from discrete sites (two of the references are even for the same site). The nature of the bed over large areas remains unknown. Therefore the capacity of the bed to store water in troughs and sediment also remains uncertain.

We edited this section to clarify discrete measurements at specific sites, and that the nature of the bed remains unknown. We have added discussion of water storage in troughs and sediments in the section above on the advice of reviewer 1.

---

## Author Response (AR2)

We thank the two reviewers for having clearly reread the manuscript and providing useful comments.
* * *
Martin Lüthi

General Comments
This manuscript has been greatly improved since its first version, and the authors carefully took into account the reviewer's comments. In my opinion, this manuscript is ready for publication now. The very minor comments below should nevertheless be taken into account. As a general remark, I think a small discussion of ice deformation vs. sliding motion would be helpful. Maybe just mentionning that dissipation from the former is very small as compared to the latter at the drill site.
-we have added text to section 3.3. to discuss this issue.

Minor Comments
185 It would be useful to reference Figure 8, explaining the designations, here.
--added--

187 "Equation (1)" (most journals put equation numbers in parentheses)
--corrected--

187 "in contact with the bed" ?
--corrected--

189 I don't fully understand Equation (4). I think geothermal heat is supplied on the whole length leading to $L_s Q_{geo}$. Also, somewhat confusingly, shear stress is transferred only on the area $L_s - L_c$, but has the magnitude $\tau_{b} = \frac{L_s}{L_s - L_c} \tau_d$ ($\tau_d$ being average driving stress). Integrated over the whole bed, the overall the dissipation still is $L_s \tau_b u_b$, as given in Equation (4). This could be made clearer.
-- We assume that the ice melt from warming of cavity water is negligible. In our calculation we assume that higher order stresses are negligible, so the shear stress at the bed is equivalent to T_d. With this in mind it is appropriate to replace T_b with T_d in Equation (4). We have made this edit, and clarified the equation with additions to the text.

193 It is not clear what kind of test this is, even if it appears twice in the sentence.
--Agreed, that was a terrible sentence. Now reworded for clarity--

294 "relieve" -> "relative" ?
--corrected--

=========

Samuel Doyle

Thank you for addressing my comments. I have three minor technical corrections:

L33 - In contrast to the statement on L32/33, Ryser et al. (2014b, J. Glac. Fig. 4b) does show acceleration into winter. Fig. 4a doesn't but then there is no data during the winter period. --this reference was added during the last revision as suggested by the other reviewer. However, we agree with this comment – there is indeed a slight hint of winter acceleration visible in the figure. We do not believe a reference here is required, nor is a debate on this figure relevant, so we have removed the reference.

L179 - Specify ice viscosity after Eq. 3 to distinguish from water viscosity. To ensure reproducibility, a value, calculation and/or justification should be given for ice viscosity as it varies non-linearly, significantly for example, between cold and temperate ice. Also effective pressure is now usually denoted N (sometimes $p_e$), with σ usually reserved for stress. I see this follows Kamb (1987) so you may choose to use σ for consistency, however, given that v has been subsituted for $u_b$ why not also use N? I note that Kamb (1987) calculates viscosity by assuming the basal shear stress equals the effective pressure. Its not clear whether the same is done here. --ice viscosity now specified and explained, and reference provided. --We initially tried to honor Kamb's notation, but you make a convincing rebuttal. We have changed this term as you suggest.

L376 - Is this 2.5 cm water or ice equivalent? Is this the regional average or a local maximum? This needs specifying. Also, the term meltwater would usually suggest a volume (or discharge) rather than a dimension. Perhaps use the term melt and then specify ice or water equivalent? --wording edited to clarify this point--